# Risk Factors Analysis of Car Door Crashes Based on Logistic Regression

**Cheng-Yong Huang** 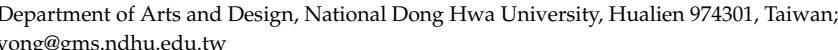

Department of Arts and Design, National Dong Hwa University, Hualien 974301, Taiwan;
yong@gms.ndhu.edu.tw

**Abstract:** Unlike door crash accidents predominantly involving bicycles in Australia, the UK, and other Western countries, cases in Taiwan are far more fatal as they usually involve motorcycles. This is due to the unique anthropogeography and transportation patterns of Taiwan, particularly the numbers of motorcycles being twice that of cars. Both path analysis and multivariate logistic regression methods were adopted in this study. The multivariate logistic regression analysis results have shown that the main risk factors causing serious injuries in door crashes include winter, morning, male motorcyclists, heavy motorcycles, and the left sides of cars. Regarding the gender differences in motorcyclists, it appears that female motorcyclists have higher door crash accident rates, while the odds of severe injury and fatality in male motorcyclists are 1.658 times greater than that of female motorcyclists. The risk factors derived from the multivariate logistic regression analysis were further discussed and analysed. It was found that the causes of serious injuries and deaths stemming from door crashes were related to the risk perception ability, reaction ability, visibility, and riding speed of the motorcyclists. Therefore, suggestions on risk management and accident prevention were proposed using advocacy through the 3E strategies of human factors engineering design.

**Keywords:** door crashes; risk factor; multivariate logistic regression; risk management; accident prevention

## 1. Introduction

Owing to the unique anthropogeography and transportation patterns of Taiwan, it has led to a rather distinctive crash type—the motorcycle door crash. In terms of anthropogeography, Taiwan is an island situated in East Asia, lying to the north-west of the Pacific. Located between the Ryukyu Islands and the Philippine Islands, it faces Eurasia across the Taiwan Strait to the west. The main island of Taiwan covers an area of 36,000 square kilometres. Taiwan's current population is about 23 million, with an average population density of 648 people per square kilometre, hence the island has a high population density. With regard to the transportation patterns of Taiwan, motorcycles are the dominant means of transport. Of the 22 million motor vehicles on the island, 14 million are motorcycles, which accounts for 64% of the overall motor vehicles. The ratio of the number of motorcycles to overall motor vehicles in Taiwan scores the highest globally [1,2]. The island is also small in area and densely populated. In order to fulfil the demand for public parking, local governments set up car parking spots along roadsides. When a driver, having parked in said car parking spot, hastily opens the door without paying attention to vehicles coming from behind, it raises the chances of striking down motorcycles coming from the rear, resulting in door crash accidents. This frequent yet unique type of traffic accident exposes victims to the dangers of being run over by oncoming cars after being knocked down. In comparison to car passengers, motorcyclists are extremely vulnerable and unprotected. Whether it is braking, sudden swerves, or other evasive maneuvers, all are relatively more difficult for motorcycles. In addition, motorcycles are less conspicuous on the road, and there is a lack of structural protection for motorcyclists

upon collisions. These factors that expose body surface areas of motorcyclists lead to their high vulnerability [3–5].

Since cycling enjoys a higher popularity in Western countries, few research studies on bicycle–car door crashes have been carried out [6–9]. Door crash accidents were first mentioned in a study by Dennerlein and Meeker [9]. Chen et al. (2018) mentioned door crash accidents in Taiwan. In their study, they suggested prosecuting illegal parking, drinking and driving, controlling the speed of motorcycles, and using the Dutch reach method to open car doors to avoid door crash accidents [10]. Additionally, Pei's [11] study investigated three kinds of bicycle accidents, namely overtaking crashes, rear-end crashes, and door crashes, respectively. A logit model was also used to analyse the risk factors [12–14]. The definition of door crashes, according to the paper, refers to accidents in which cyclists are hit by the doors of cars parked on the roadside. It was mentioned that cyclists experience injury from flipping over upon striking the car door, and a second injury upon ground collision, causing serious damage, especially for cyclists not wearing safety helmets. The study also mentioned that most cycle lanes are set up in the "door zone" of parked cars [15]. To put this into perspective, a car door extends about 90–105 cm outwards when it is open; cycle lanes span just a little over 90–105 cm. Even though cyclists usually ride in the middle of these designated lanes, this undoubtedly elevates the risk of a bicycle crashing into an open door. Whether or not cyclists ride on cycle lanes, they are often instructed to keep a door's distance from cars parked by the roadside. This enhances the visual field of drivers parked on the roadside and the conspicuity of cyclists. The study also found that for fixed parameters, in addition to cycle lanes being right next to parking spots along kerbs, passengers frequently getting in and out of taxis, female cyclists and female car drivers are also commonly involved in door crash accidents [16]. For random parameters, it was found that one-way streets, older cyclists, and daytime also result in higher rates of door crashes.

In a study by Johnson et al. [6] on bicycle–car door crash accidents, door crashes were defined as the potential collision when the door of a car parked parallel to the roadside was opened, entering the cycle lanes of approaching cyclists. It was also mentioned in the article that cyclists involved in door crashes often suffer severe injuries, but the characteristics and harm that stems from such collisions were unknown. Four cyclist deaths in door crash accidents were recorded in Victoria, Australia from 1989 to 2012. The study is based on the analysis of police traffic accident reports of door crash accidents and hospital reports. It shows that male cyclists, aged 18 years and above, on roads with a speed limit of 60 km/h or more, are more prone to accidents involving bicycle–car door crashes.

Based on the review of the research above, the risk factors of bicycle–car door crashes are clear and the difference in risk factors of bicycle–car door crashes among various countries was made apparent. This is due to different data sources; that is especially the case regarding the gender of cyclists. According to Pei's [11] research results in the United Kingdom, door crashes occurred more often with females. Conversely, across the world in Australia, the research results of Johnson et al. [6] showed that door crashes occurred more with males. The stark contrast is obvious. The difference between the two calls for further clarification and verification in this study. The distinct traffic patterns of Taiwan, as opposed to the rest of the world, have resulted in rare cases of bicycle–car door crashes; on the contrary, an abundance of motorcycle–car door crash accidents. This study investigates the risk factors causing critical and lethal damage to motorcyclists involved in door crash accidents. Presently, there are no relevant research investigations and analyses at home and abroad, and this topic requires clear injury data to enable statistical analysis. The data source of this research is drawn from the verdicts of Taiwan District Courts, with a total of 491 cases from 1995 to 2020.

This study focuses on the statistical investigation and analysis of the risk factors causing serious injuries and deaths by motorcycle–car door crash accidents in Taiwan. The civil verdicts of district courts across various counties and cities in Taiwan that have clear data on injuries and deaths serve as the source data. The content of verdicts classifies

risk factors into season, time zone, day and night, off-peak, gender of the victim, gender of the perpetrator, type of motorcycle, side of the car, etc.; the injuries are divided into binary categories of minor injuries and grave casualties. Using binary logistic regression, an odds ratio calculation of the risk factors on severe casualties with statistically significant differences was made. The following chapters will elaborate in detail the data sources and classification status, statistical methods used, statistical analysis results, discussion, and suggestions on risk management proposed using advocacy through the 3E (education, enforcement and engineering) strategies of human factors engineering design [17].

## 2. Methodology and Data Sources

### 2.1. Logistic Regression Analysis

Regression analysis technology was used to understand whether two or more variables are correlated, the direction and the strength of said correlation, and to establish a mathematical model to predict the variables. In traditional regression analysis, a continuous variable is required to be the dependent variable when making model predictions. When the dependent variable is a categorical variable, a log-linear model is required. Logistic regression was one of the methods used to predict the categorical dependent variable. Logistic regression analysis is suitable for binary category data with dependent variables. It was adopted in this study to produce analytic results on risk factors causing severe injuries and deaths in door crash accidents [18]. Similar to that of Pei [11], its main objective is to predict the regression analysis of the probability of an event.

In this study, the collected data on door crashes were facts that have occurred and categorised as a binary nominal dependent variable [19]. Assuming Y is the dependent variable, it represents both the scenarios of minor injuries or severe injuries including death. In this case, assume that minor injuries are zero, serious injuries and deaths are one, and $X = (X_1, X_2, X_3, \ldots X_n)$ are independent and known observations, that is, independent variables, which can be continuous variables or categorical variables; $P(x)$ indicates the probability of success. This probability value varies with X, but is between zero and one. The probability of serious injury and death in an accident can then be expressed as: $P(X) = e^{f(x)}/1 + e^{f(x)}$, the probability of being slightly injured in an accident can be expressed as: $1 - P(X) = 1/1 + e^{f(x)}$, and so the odds ratio can be expressed as: odds $= P(X)/1 - P(X) = e^{f(x)}$, this is the intensity of the probability of serious injury and death in an accident relative to the probability of minor injury. Therefore, multivariate logistic regression is adopted for predicting the probability of the occurrence of serious injuries including deaths. In SPSS multivariate logistic regression, variables in the independent variables category are converted to dummy codes to designate a category as a reference group. The Exp(B) calculated by multivariate logistic regression in SPSS represents the odds ratio, and is the multiple of the probability of occurrence for a variable compared with the reference group [20].

The goodness of fit test of the logistic regression model can be verified via log likelihood (LL function). The significance test of the entire logistic regression includes the overall model verification and the individual parameter test. The goodness of fit of the regression model can be verified via Pearson $X^2$, Hosmer–Lemeshow, and $R^2$-like values. Hosmer–Lemeshow was used in this study to verify the model suitability. Assume that the $H_0$: Logistic regression model was fit, then with the Chi-square test, significance greater than 0.05 (significant level) means that the $H_0$ null hypothesis cannot be rejected, indicating that the logistic regression model holds a certain degree of model predictability [21]; while the significance test of individual parameters is verified by the Wald test, significance (Sig) lesser than 0.05 (significant level) represents a significant difference.

### 2.2. Source of Information and Categorisation

The main source of information of this study was the verdict database query system of the Judicial Court (5 September 2020, Retrieved from https://law.judicial.gov.tw/FJUD/default_AD.aspx). The verdicts included those of the Taiwan Supreme Court, High Courts,

and the District Courts after 1995. In this study, the scope of the investigated data includes the civil verdicts from 15 District Courts, with the term "car door" as the keyword for query, and a total of 491 cases from April 1995 to January 2020 were found. The verdicts contain detailed descriptions about the injuries of the motorcyclists, which were the summary of hospital diagnosis, and therefore, judgement on the injury of involved motorcyclists can be made. Injuries that one can fully recover from such as abrasions, bruises, minor fractures, etc., are regarded as minor injuries. Injuries such as severe internal injury, intracranial hemorrhage, severe fracture, amputation, disability, coma, and death are considered serious injuries. Among the 491 cases collected, 188 were cases of serious injuries, while minor injuries made up 303 cases. In order to classify the collected data, we must first understand the risk factors that cause serious injuries in accidents, before performing appropriate classification statistics for the textual data within the verdict. Keay and Simmonds [22] discussed the impact of rain and snow on traffic accidents in Melbourne, Australia, and found that higher rainfall had the greatest impact on accidents. The average number of cases was 19% higher on rainy days than the overall average. In autumn particularly, the rainfall effect on accidents in Australia yields a greater significance.

In addition to a comprehensive description of each motorcyclist's injury, the court verdicts contain information such as the date, time, location, car perpetrator, injured victim, description of the accident, type of motorcycle, etc. Therefore, it is necessary to record and classify these statistics individually. When the verdict database query system was established in 1995, gender was not disclosed in the personal data, and names were hidden. However, 2010 saw a change stipulating that the name of the natural person in verdicts would now be disclosed. Therefore, the gender of the injured motorcyclists and the perpetrator could now be determined from their names. There were a total of 308 such cases recorded from 2008 to 2020.

After a preliminary processing of the text data of 491 door crash cases verdicts from 1996 to 2020, the data call for further conversion and categorisation. Dates are divided into seasons, i.e., spring, summer, autumn, and winter. Located on the Tropic of Cancer, Taiwan exhibits four distinct seasons. Spring spans from March to May, summer is usually June to August, September to November brings in autumn, and winter is generally from December to February. Time information was converted into time periods, namely morning, afternoon, and night. Due to the latitude, Taiwan expects sunrise at around 6 a.m. in the morning and sunset at around 7 p.m. in the evening. Therefore, the morning hours are from 6 a.m. to 12 p.m., the afternoon hours are from 1 p.m. to 7 p.m., and the night hours are from 7 p.m. to 6 a.m. The off-peak variables are peak hours from 7 a.m. to 9 a.m. in the morning, lunch break hours from 11 a.m. to 1 p.m., and 5 p.m. to 7 p.m. in the evening after work, respectively. The rest are considered off-peak hours. The cars involved were split into accidents occurring on the left or the right of the vehicle. Vehicles in Taiwan drive on the right side of the road and therefore most of the door crash accidents occurred on the left side of a stationary car. The few cases where the accidents took place on the right side of cars were caused by the cars parking too far from the roadside. The genders of injured motorcyclists and perpetrators were divided into male and female in the 308 cases, and were determined from their names following the disclosure of names in verdicts after 2008. The motorcycles were categorised into light motorcycles and heavy motorcycles by engine size. According to Penumaka et al. [23], the size of the engine serves as a classification standard for two-wheel-powered motor vehicles. Taiwan's approach towards motorcycle management has long used engine size to distinguish between motorcycles. Motorcycles with engine displacements of lower than 50 cc are considered light motorcycles, and engine displacements of higher than 50 cc are considered heavy motorcycles. In contrast to light motorcycles, heavy motorcycles are faster, and are able to travel a further distance. Finally, the matter of most importance—levels of injury. Divided into minor and severe injuries, they are mainly gauged from the textual description of injury in the verdicts and the amount of compensation given. Since the acquisition of gender data for cases after 2008,

the data are organised for classification statistics into 491 pieces of datum for the long-term period from 1995 to 2020 and 308 pieces of data for the short-term period from 2008 to 2020.

## 3. Statistical Results

### 3.1. Descriptive Statistics

Table 1 shows the proportion of the number of classifications in the long-term period from April 1995 to January 2020 and in the short-term period from February 2008 to January 2020. The statistical results regarding the long-term period show that the proportion of minor injuries, at 61.7%, was higher than the proportion of serious injuries and death, at 38.3%. When it came to the seasons, accidents in spring, with a proportion of 28.7%, were higher than that of the remaining three seasons, with summer at 22.2%, autumn at 22.8%, and winter at 26.3%. In terms of time period, the percentages of 45.2% in the morning and 39.7% in the afternoon were much higher than the 15.1% in the evening. The same results were also presented in the day–night data. The proportion of accident occurrences in the daytime reached a staggering 84.9%, far greater than 15.1% in the night. Regarding the daily traffic hours, the proportion of accidents during rush hour was 63.3%, higher than that of off-peak traffic at 36.7%. As for the car sides, the proportion of accidents occurring on the left was as high as 87.0%, far exceeding 13.0% on the right. Concerning the motorcycle engine however, the proportion of accidents involving light motorcycles, 70.1%, was much higher than that of heavy motorcycles, at 29.9%.

**Table 1.** The occurrence ratio of each variable in the long-term and short-term.

| Selected Variable | | 1995~2020 (Total 491 Cases) | | | 2008~2020 (Total 308 Cases) | | |
|---|---|---|---|---|---|---|---|
| | | Minor Injury | Serious Injury | Total Number | Minor Injury | Serious Injury | Total Number |
| | | 303 (61.7%) | 188 (38.3%) | 491 (100%) | 196 (63.6%) | 112 (36.4%) | 308 (100%) |
| Season | Spring | 86 (17.5%) | 55 (11.2%) | 141 (28.7%) | 60 (19.5%) | 30 (9.7%) | 90 (29.2%) |
| | Summer | 73 (14.9%) | 36 (7.3%) | 109 (22.2%) | 47 (15.3%) | 20 (6.5%) | 67 (21.8%) |
| | Autumn | 76 (15.5%) | 36 (7.3%) | 112 (22.8%) | 47 (15.9%) | 23 (7.5%) | 70 (22.7%) |
| | Winter | 68 (13.8%) | 61 (12.4%) | 129 (26.3%) | 42 (13.6%) | 39 (12.7%) | 81 (26.3%) |
| Time period | Morning | 130 (26.5%) | 92 (18.7%) | 222 (45.2%) | 78 (25.3%) | 57 (18.5%) | 135 (43.8%) |
| | Afternoon | 118 (24.0%) | 77 (15.7%) | 195 (39.7%) | 86 (27.9%) | 46 (14.9%) | 132 (42.9%) |
| | Night | 55 (11.2%) | 19 (3.9%) | 74 (15.1%) | 32 (10.4%) | 9 (2.9%) | 41 (13.3%) |
| Day–Night | Day | 248 (50.5%) | 169 (34.4%) | 417 (84.9%) | 164 (53.2%) | 103 (33.4%) | 267 (86.7%) |
| | Night | 55 (11.2%) | 19 (3.9%) | 74 (15.1%) | 32 (10.4%) | 9 (2.9%) | 41 (13.3%) |
| Traffic hours | Peak | 200 (40.7%) | 111 (22.6%) | 311 (63.3%) | 132 (42.9%) | 71 (23.1%) | 203 (65.9%) |
| | Off-peak | 103 (21.0%) | 77 (15.7%) | 180 (36.7%) | 64 (20.8%) | 41 (13.3%) | 105 (34.1%) |
| Motorcyclist Gender | Male | NA | NA | NA | 72 (23.4%) | 57 (18.5%) | 129 (41.9%) |
| | Female | NA | NA | NA | 124 (40.3%) | 55 (17.9%) | 179 (58.1%) |
| Perpetrator Gender | Male | NA | NA | NA | 143 (46.4%) | 80 (26.0%) | 223 (72.4%) |
| | Female | NA | NA | NA | 53 (17.2%) | 32 (10.4%) | 85 (27.6%) |
| Engine Size | Light | 110 (22.4%) | 37 (7.5%) | 147 (29.9%) | 57 (18.5%) | 15 (4.9%) | 72 (23.4%) |
| | Heavy | 193 (39.3%) | 151 (30.8%) | 344 (70.1%) | 139 (45.1%) | 97 (31.5%) | 236 (76.6%) |
| Car side | Left | 254 (51.7%) | 173 (35.2%) | 427 (87.0%) | 164 (53.2%) | 100 (32.5%) | 264 (85.7%) |
| | Right | 49 (10.0%) | 15 (3.1%) | 64 (13.0%) | 32 (10.4%) | 12 (3.9%) | 44 (14.3%) |

Note: The denominator of the long-term percentage is the total number of long-term cases, 491, and the denominator of the short-term percentage is the total number of short-term cases, 308.

The short-term statistical results on the other hand, show that the proportion of minor injuries, at 63.6%, was higher than that of serious injuries and death at 36.4%. In terms of the four seasons, accidents in spring, with a proportion of 29.2%, were higher than that of the remaining three seasons, with summer at 21.8%, autumn at 22.7%, and winter at 26.3%. Regarding the time period, the proportion of accidents at 43.8% in the morning and 42.9% in the afternoon were much higher than the 13.3% in the evening. When it came to the day–night results, the proportion of occurrences in the daytime reached a staggering 86.7%, far greater than the 13.3% in the night. As for the daily traffic hours, the proportion of rush hour traffic accidents was 65.9%, higher than that of off-peak traffic at 34.1%. With regards to the sides of the car, the proportion of accident occurrences on the left was as high as 85.7%, far exceeding 14.3% on the right. Regarding the gender of injured motorcyclists, the proportion of females was 58.1%, higher than that of men, at 41.9%. When it came to the gender of the perpetrators however, the proportion of males causing accidents was 72.4%, much higher than that of females, at 27.6%. As for the motorcycle engine, the proportion of accidents involving light motorcycles, coming at 76.6%, was much higher than that of heavy motorcycles at 23.4%.

### 3.2. Risk Factors of Severe Casualties Caused by Door Crash Accidents

The risk factors of severe casualties caused by door crash accidents were analysed using multivariate logistic regression. In this study, the collected data on door crashes was categorised as a binary nominal dependent variable, i.e., minor injury (0) and serious injury (1). At this stage, due to the change in the original registration method of the Taiwan District Court that allowed the name of the natural person to be disclosed in verdicts made since 2008, the gender of the injured motorcyclists can be determined from their names. Thus, it will be divided into two parts of multivariate logistic regression. The first part, which excludes the gender of injured motorcyclists, is a long-term view of a total of 491 cases. The second part, with a short-term record of 308 cases, includes the gender of the injured motorcyclists. The multivariate logistic analysis of the risk factors of severe casualties caused by door crash accidents concerning both long-term and short-term cases, is summarised as shown in Table 2, where Exp(B) represents the odds ratio multiple, and Sig. represents significance. At a confidence level of above 95%, the long-term model includes independent variables that exhibit significant differences in seasons, day and night, motorcycle types, and the sides of the car involved. In terms of the goodness of fit of the long-term model, the goodness of fit test by Nagelkerke $R^2 = 0.097$, Hosmer, and Lemeshow assumes $H_0$: Logistic regression model is fit, the test statistic chi-square value is 2.272, and the significant probability is 0.943 > 0.05 (significant level), so the null hypothesis cannot be rejected, indicating that the model has a certain predictability and the goodness of fit of the model is acceptable. At a confidence level of below 95%, the short-term model includes independent variables that exhibit significant differences in seasons, time periods, motorcycle types, and the gender of the injured motorcyclists. In terms of the goodness of fit of the short-term model, the goodness of fit test by Nagelkerke $R^2 = 0.126$, Hosmer, and Lemeshow assumes $H_0$: Logistic regression model is fit, the test statistic chi-square value is 5.696, and the significant probability is 0.618 > 0.05 (significant level), so the null hypothesis cannot be rejected, indicating that the model has a certain predictability and the goodness of fit of the model is acceptable.

From a long-term perspective, there is no significant difference in severe casualties between winter and spring, but a striking difference arises when summer and autumn are put into comparison. The odds ratio of severe casualties in winter is 1/0.520 = 1.923 times that of summer, which is 1/0.445 = 2.247 times that of autumn. Regarding the time period, the day and the night show a clear comparison. The odds ratio of severe casualties caused by door crashes during the day are 1.908 times that of the night. In terms of the motorcycle type, the odds ratio of a heavy motorcycle involved in a door crash resulting in serious injuries and deaths are 2.341 times that of a light motorcycle. The odds ratio of a door crash

accident involving serious injuries and deaths on the left side are 2.067 times that of the right side.

**Table 2.** Summary of risk factors of serious injuries by multivariate logistic regression.

| Risk Factor | 1995~2020 (Total 491) | | | | | 2008~2020 (Total 308) | | | | |
|---|---|---|---|---|---|---|---|---|---|---|
| | B | S.E. | Wals | Sig. | Exp(B) | B | S.E. | Wals | Sig. | Exp(B) |
| Winter | | | 9.862 | 0.020 | | | | 8.657 | 0.034 | |
| Spring/Winter | −0.440 | 0.256 | 2.942 | 0.086 | 0.644 | −0.779 | 0.332 | 5.504 | 0.019 | 0.459 * |
| Summer/Winter | −0.655 | 0.278 | 5.544 | 0.019 | 0.520 * | −0.880 | 0.363 | 5.886 | 0.015 | 0.415 * |
| Autumn/Winter | −0.810 | 0.279 | 8.415 | 0.004 | 0.445 ** | −0.807 | 0.356 | 5.132 | 0.023 | 0.446 * |
| Morning | | | | | | | | 6.509 | 0.039 | |
| Afternoon/Morning | | | | | | −0.352 | 0.265 | 1.767 | 0.184 | 0.703 |
| Night/Morning | | | | | | −1.055 | 0.428 | 6.088 | 0.014 | 0.348 * |
| Day/Night | 0.646 | 0.296 | 4.779 | 0.029 | 1.908 * | | | | | |
| Male/Female Motorcyclist | | | | | | 0.506 | 0.253 | 3.987 | 0.046 | 1.658 * |
| Heavy/Light | 0.851 | 0.225 | 14.343 | 0.000 | 2.341 *** | 1.016 | 0.334 | 9.221 | 0.002 | 2.761 ** |
| Left/Right | 0.726 | 0.324 | 5.015 | 0.025 | 2.067 * | | | | | |
| Constant | −1.845 | 0.431 | 18.336 | 0.000 | 0.158 *** | −0.724 | 0.370 | 3.835 | 0.050 | 0.485 |

Note: Exp (B) stands for odds ratio multiples; sig. stands for significance, * stands for statistically significant differences; ** stands for statistically highly significant differences; *** stands for statistically extremely significant differences.

From a short-term perspective, there is a significant difference between winter and spring, and between summer and autumn. The odds ratio of severe casualties in winter is 1/0.459 = 2.179 times that of spring, it is 1/0.415 = 2.409 times that of summer, which is also1/0.446 = 2.242 times that of autumn. Regarding the time period, the odds ratio of severe casualties is highest in the morning. Accidents occurring in the morning is 1/0.703 = 1.422 times that of accidents happening in the afternoon, and is 1/0.348 = 2.874 times that of accidents happening in the evening. In terms of the gender of injured motorcyclists, the odds of males suffering from serious injuries and death are higher than women, with an odds ratio of 1.658 times. In terms of the motorcycle type, the odds of a heavy motorcycle involved in a door crash resulting in serious injuries and deaths are 2.761 times that of a light motorcycle.

Considering both the long-term and short-term risk factors of severe casualties caused by door crash accidents, it is clear that when it comes to the seasons, winter has the highest odds of accidents and deaths. An accident occurring during daytime also accounts for higher odds of severe casualties in comparison to night-time, the highest being in the morning. Male motorcyclists are more likely to suffer serious injuries than female motorcycle riders in accidents. Door crash accidents involving heavy motorcycles are more likely to yield severe casualties than those involving light motorcycles. Lastly, in the unfortunate event of a door crash accident, those occurring on the left side of the car, as compared to the right side, produce deeper damage.

## 4. Discussion

Having completed the multivariate logistic regression analysis of the risk factors of door crash accidents, it was found that the factors under the risk factors such as season, time period, day and night, gender of the injured motorcyclist, type of motorcycle, and the side of the car have an immense impact on the severity of casualties.

The proportion of severe injuries and deaths however, is the highest in winter, with 12.4% in the long-term cases and 12.7% in the short-term cases. Compared with other seasons, the odds of severe injury and death in winter are 2.179 times that of spring,

2.409 times that of summer and 2.242 times that of autumn. It can be gleaned from the data above, that winter and spring have the most occurrences of accidents and the highest probability of causing serious injuries and deaths. Hence, the total number of door crash accidents and the statistics of the number of severe casualties in each month over the long-term period are indicated in Figure 1. The results reveal that door crash accidents are concentrated in the period from January to May, and peak during January, April, and May. This brings about the highest number of accidents and cases of serious injuries and deaths, the phenomenon of which is related to the climate in Taiwan. According to research by Hjelkrem and Ryeng [24], winter and rainy days deteriorate the risk perception of car drivers. January brings the coldest month to the island, and April to May is the rainy season. Therefore, winter and rainy days become the main contributors that lead to door crash accidents and severe casualties.

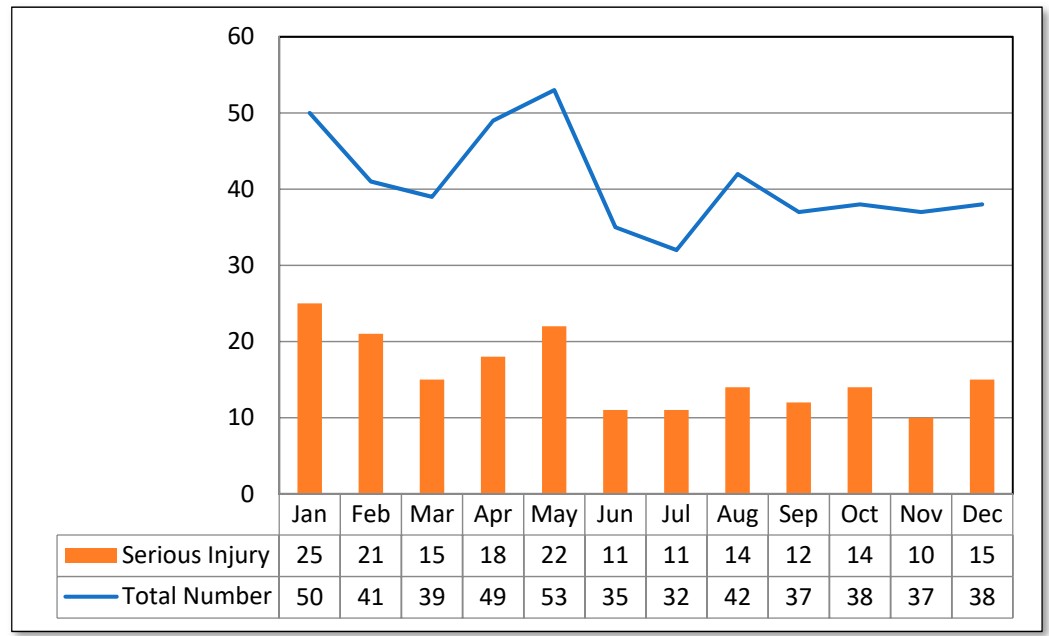

**Figure 1.** The total number of door crashes and the number of serious injuries in each month during the long-term period.

The time period can be divided into time zones consisting of morning, afternoon, and evening, day–night, and traffic hours for discussion. As for the short-term period, the highest number of accidents was 43.8% in the morning, followed by 42.9% in the afternoon, and the lowest in the evening, at only 13.3%. With the highest odds of casualties in the morning (long-term 18.7%, short-term 18.5%), followed by the afternoon, and the lowest in the evening, no significant difference is shown between the long and short-term periods. There is however, a striking difference in the odds of casualties for the short-term in multivariate logistic regression, with the morning being 1/0.703 = 1.422 times that of the afternoon, and being 1/0.348 = 2.874 times that of the evening. In terms of day and night, the proportion of daytime accidents is remarkably high, at 84.9% for long-term periods and 86.7% for short-term periods. Night-time on the other hand is proportionally much lower. The proportion of serious injuries and deaths is similar to the number of occurrences, being much higher during the day than at night. In the case of serious injuries and deaths, the odds ratio during the day is 1.908 times that of at night. Such findings are consistent with the results of Jaber et al. [25]. Finally, regarding the traffic hours, 63.3% of the accidents in the long-term period occurred during the day, and with the short-term period it was 65.9%; the remaining percentage of accidents occurred in the off-peak hours. However, the odds of severe casualties were not significantly different from the peak hours. The number of occurrences per hour and the number of severe casualties in the short-term period from 2008 to 2020 were hence further sorted out, as shown in Figure 2. It was made

apparent from the number of occurrences, with 32 cases, that 5 p.m. marks the peak of door crash accidents. Additionally, the time periods with considerable cases of severe casualties were 8 a.m., 10 a.m. to 12 p.m., and 5 p.m., of which 8 a.m. and 5 p.m. being the peak hours of getting to work in the morning and getting off work in the afternoon, respectively. The resulting greater traffic flow therefore, comes as no surprise. What is noteworthy is that from 11 a.m. to 12 p.m., the ratio of the number of severe casualties divided by the total number of occurrences in this period exceeds 50%. This may be due to the fact that the traffic flow at noon, with people going out for business and lunch, is less than that of people travelling to and from work. Therefore, the ratio of severe casualties (the number of severe casualties in the current month divided by the total number of accidents in the month) caused by faster motorcycles is also considerably higher.

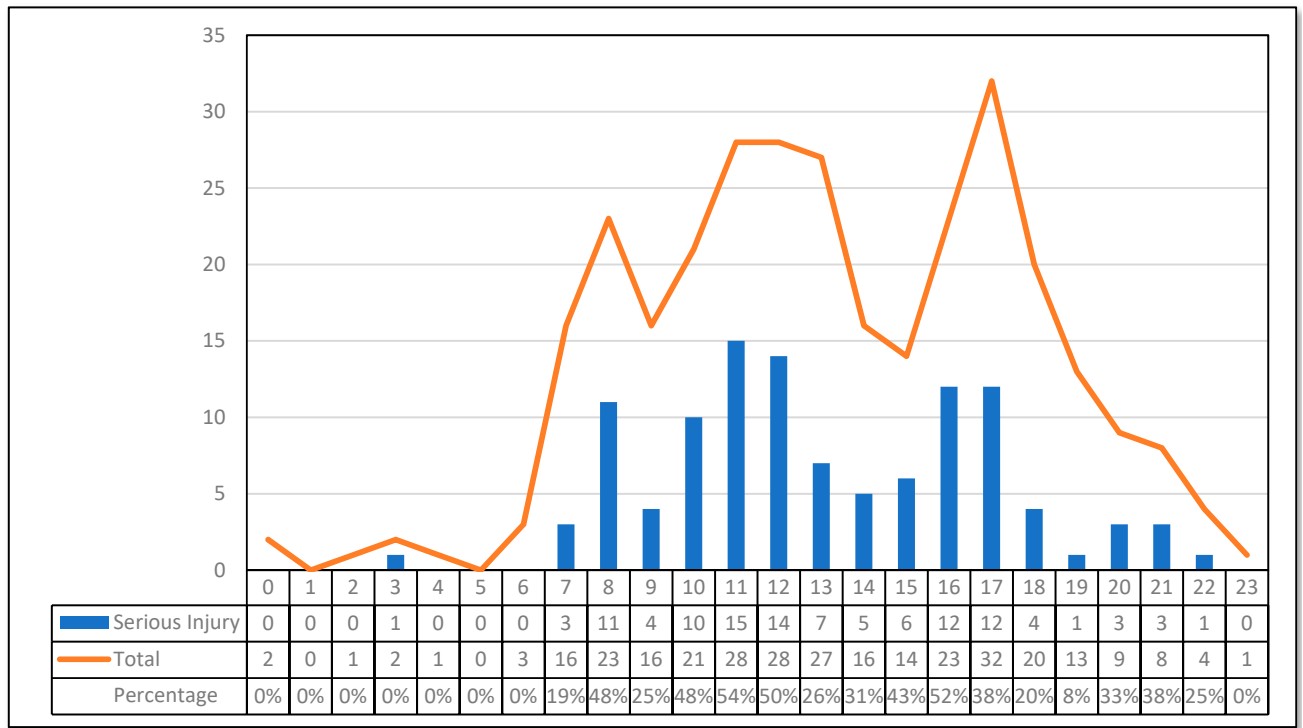

**Figure 2.** The total number of door crashes and the number of serious injuries during each hour in the short-term period.

In terms of gender, it can be split into the gender of the perpetrator and of the injured victim. According to official public information (28 August 2020, Retrieved from http://www.gender.ey.gov.tw/), in 2018, the gender ratio of automobile drivers with a driver's licence was 67.8% male and 32.2% female. With the short-term data, the proportion of male perpetrators is 72.4%, which is much higher than the proportion of female perpetrators, at 27.6%. Male perpetrators are also responsible for 26.0% of the casualties caused compared to female perpetrators at 10.4%. The multivariate logistic regression calculations however, produced no significant differences between male and female perpetrators in door crash accidents, meaning whether it be a male or female perpetrator, the odds of causing serious injuries to the motorcyclist are rather similar. This is close to the gender ratio of the car drivers. Based on the gender of affected motorcyclists, females with a percentage of 58.1% are more likely than males, at 41.9%, to be caught in door crash accidents. Male motorcyclists are also responsible for 18.5% of the casualties caused compared to female motorcyclists at 17.9%. According to official public information, the gender ratio of having a motorcycle driver's licence is 54% male and 46% female. As a result of the lower proportion of male motorcyclists, it was found through multivariate logistic regression that the odds of severe injuries and death for male motorcyclists are 1.658 times greater than that of female motorcyclists. The greater odds of serious injury for male motorcycle drivers require attention.

In an experiment conducted by Scrimgeour et al. [26], there was no difference seen in risk perception between genders, but in the experiment conducted by Fitch et al. [27], male drivers, compared with female drivers, had a relatively faster perception time and applied a relatively greater force on the brake pedal, generating a larger deceleration and shorter stopping distance. Men had a slightly higher intention to speed than women. The impact of sensation seeking and injunctive norms was stronger on men's intention to speed, whereas self-description variables had a greater impact on women's intention. Speeding intention increased with driving experience [28,29]. Due to their slower response times, female motorcyclists are more often involved in door crash accidents, but because of their lower speed, female motorcyclists are less likely to suffer grave injuries. Male motorcyclists, on the other hand, can respond faster and are less likely to suffer a door crash accident, but because of their faster speeds, victims suffer more serious injuries. According to official public information (28 August 2020, Retrieved from http://www/ris/gov/tw/), in 2016 there were a total of 13,656,001 motorcycles in Taiwan, of which 11,818,526 (86.5%) were heavy motorcycles and 1,837,475 (13.5%) were light motorcycles. As for the type of motorcycles, in the long-term period, heavy motorcycles accounted for 70.1% of the door crashes as compared with only 29.9% of the light motorcycles. Similar results were shown in the short-term data, with 76.6% of heavy motorcycles being involved in door crash accidents, greater than the 23.4% of light motorcycles. In terms of severe injuries and fatalities, in the long-term, the proportion of heavy motorcycles suffering severe injuries and fatalities was 30.8%, higher than that of light motorcycles at 7.5%, while in the short-term, the proportion of heavy motorcycles suffering serious injury or death was 31.5%, higher than that of light motorcycles at 4.9%. Based on the results of the multivariate logistic regression model, it was found that, in the long term, the odds of heavy motorcycles resulting in severe injuries and deaths are 2.341 times higher than those of light motorcycles. As for the short-term, heavy motorcycles resulting in severe injuries and deaths were 2.761 times that of light motorcycles. The engines of heavy motorcycles provide higher displacements and faster riding speeds that are related to the riders as well. Therefore, a cross table of short-term data regarding gender, type of motorcycle and injuries was listed to examine the odds ratio, as shown in Table 3.

**Table 3.** Cross table of the ratio of gender to the number of occurrences by motorcycle type and the odds ratio of severe casualties.

| | Minor Injury | Severe Injury Including Death | Total | Multiples of the Odds Ratio |
|---|---|---|---|---|
| Males riding heavy motorcycles (MH) | 62 (20.1%) | 49 (15.9%) | 111 (36.0%) | 1.000 (MH/MH) |
| Females riding heavy motorcycles (FH) | 78 (25.3%) | 48 (15.6%) | 126 (40.9%) | 0.778 (FH/MH) |
| Males riding light motorcycles (ML) | 10 (3.2%) | 8 (2.6%) | 18 (5.8%) | 1.012 (ML/MH) |
| Females riding light motorcycles (FL) | 46 (14.9%) | 7 (2.3%) | 53 (17.2%) | 0.192 (FL/MH) |
| Total | 196 (63.6%) | 112 (36.4%) | 308 (100%) | |

The results show that males riding heavy and light motorcycles come highest in the odds ratio of severe casualties, although females riding heavy motorcycles come next. Finally, females riding light motorcycles come last. In the above results, a clear correlation could be seen between the risk factors of door crash accidents, serious injuries, and death, as well as gender and type of motorcycle. With males riding motorcycles being the fastest, and females riding light motorcycles being the slowest, the degree of casualties caused is therefore different.

Regarding the side of the car, it appears that the proportion of door crash accidents on the left side (driver's side) was as high as 87.0%, considerably higher than 13.0% on

the right. The proportion of serious injuries and deaths on the left side, at 35.2%, was also greater than the 3.1% on the right. The multivariate logistic regression reveals significant differences in the occurrence of car side risk factors. As far as the long-term results are concerned, the odds ratio of severe casualties resulting from door crashes on the left side is 2.067 times greater than on the right. Therefore, it is imperative that drivers and passengers alike pay close attention to the moving motorcycles behind when they exit the vehicle, as the impact from the left side will most likely result in the motorcyclist suffering severe casualties.

## 5. Conclusions

Following the discussion above, it appears that serious injury and death resulting from door crash accidents were also impacted by season, time zone, day and night, gender of the motorcyclist, type of motorcycle, and side of the car. The factors were winter, morning, daytime, male motorcyclist, heavy motorcycle, and the left side of the car. Further analysis suggests that motorcyclists and drivers have poorer risk perception during winter and rainy days. Moreover, the speed of motorcyclists during the rush hour at noon is faster than that of the morning and evening commuting hours. As well as gender causing differences in the ability to respond to changes, the less conspicuous nature of motorcycles plays a role. Males riding heavy motorcycles and on the left side of cars are also potential causes of serious injuries and deaths resulting from door crashes. Therefore, refining the risk perception ability of car drivers and motorcyclists, boosting the response ability of motorcyclists to accidents, improving the conspicuity of motorcycles, and reducing the speed of motorcyclists are the key to preventing serious injuries and deaths caused by door crash accidents. With the 3E strategies of Education, Enforcement, and Engineering, the following offers suggestions on risk management.

As males and heavy motorcycles tend to travel faster, the role of education and advocacy is particularly distinct during the license examination of heavy motorcycles. Candidates, especially males, should be taught to slow down when approaching cars on the roadside, and to be mindful of whether or not there are passengers exiting the vehicle. It is best to ride at a distance of more than one metre from a parked vehicle on the roadside. It is common in Europe to use the "Dutch Reach" measures to prevent bicycle–door collisions because the risks of serious injury and death are relatively high in door crashes on the left side of the car. Presently, the "Dutch Reach" two-stage door opening has also been adopted by Taiwan as one of the topics in the car license examination. In addition to driving, it is also recommended to promote the two-stage door opening method when the passengers on the left get out of the car to ensure that there is no motorcycle coming from behind before opening the door [10].

In terms of the implementation of regulations, the speed limit of motorcycles should be reduced during the peak hours at noon and on the roadside where there are car parking spots. It is also proposed that the legislature require motorcycles to have their headlights on during the day to increase their visibility, and mandate that all cars have child safety locks on the rear left doors so that passengers in the rear left seats are unable to open the door and leave the car. Finally, it is necessary to reinforce the practice of two-stage door opening in cars, and the ban on motorcycle speeding, so as to prevent door crash accidents.

The risk of serious injuries and death due to accidents on the left side of the car is far greater than that on the right side. Therefore, in terms of engineering design, it is recommended to attach a ready-to-get-out indicator light on the left side mirror, which indicates that someone is getting out of the car, or to change the colour of the rear lights to warn, just as the third brake light does so. A study and research conducted by Kahane [30] discovered that adding a third brake light to a vehicle effectively reduces 17% of car accidents involving a rear vehicle. Sivak et al. [31] pointed out that the third brake light prepares the test subject mentally by putting one in anticipation, which reduces the differential reaction time by 0.18 when braking. The car's side mirror is equipped with a ready-to-get-out indicator light, which alerts passing motorcyclists to the fact that the

driver or the passengers are getting ready to get out. The ready-to-get-out indicator light is activated when the car is stationary and shifted to the P gear, only turning off when the driver turns off the engine and locks the door, or when the driver shifts the gear from P gear to R gear, N gear, or D gear. Passing motorcyclists can then anticipate that there are drivers or passengers in the car parked by the side of the road, therefore slowing down accordingly and keeping a safe distance from the car to prevent potential door crash accidents.

The main contribution to this research is the collection of 491 civil verdicts of the Taiwan District Court from 1995 to 2020 regarding door crash accidents, which are divided into minor injuries and severe casualties, and the use of multivariate logistic regression to explore the factors causing serious injuries and death risk in door crash accidents. Following statistical analysis and discussion, according to the 3E strategy of education, law enforcement, and engineering, recommendations for risk management in accident prevention were proposed. This includes the human factors engineering concept of preparing ready-to-get-out indicator lights, so motorcyclists are alerted to the possibility that someone is getting out of a parked car, hence avoiding the car or slowing down. Furthermore, this study has made important findings regarding the gender of the injured victims. In other words, female motorcyclists are more inclined to suffer door crash accidents, and male motorcyclists tend to suffer more severe injuries.

This research, at present, has initially figured out the risk factors of serious injury and death in door crash accidents. It is recommended that future research examine other risk factors overlooked in this study, such as the speed of the motorcycle, the safety distance between the car and the motorcycle, the behaviour of the car driver opening the door, and so on. However, the data on these factors are inaccessible from court verdicts. Experiments and tests must be conducted in conjunction with other methods, such as human engineering and behaviour observation. Only then can we have a comprehensive understanding of the risk factors of door crash accidents and work to prevent such accidents through risk management and prevention.

**Funding:** This research received no external funding.

**Institutional Review Board Statement:** Not applicable.

**Informed Consent Statement:** Not applicable.

**Data Availability Statement:** The raw data that support the findings of this study are openly available in Taiwan Judicial Yuan at https://law.judicial.gov.tw/FJUD/default_AD.aspx.

**Conflicts of Interest:** The author declares no conflict of interest.

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
