# Peer review of "Risk Factors Analysis of Car Door Crashes Based on Logistic Regression"

_sustainability, doi:10.3390/su131810423_

Round 1

Reviewer 1 Report

The manuscript focuses on car door accidents involving motorcyclists based on data about crashes. The paper is well written. Minor comments were made.

Minor remarks

Remark 1. Please remove the instructions for authors between line 149 and 156.

Remark 2.  ’It is made apparent from the perspective of the seasons that the odds of door crash 316 accidents are the highest in spring, with analysis of long-term cases being 30.6% and 33.8% 317 in short-term cases.’

This finding is not justified because the seasonal traffic volumes are not presented. Greater traffic volume may increase the total number of accidents, but not the odds. Therefore, the findings may focus on injure severity.

Remark 3. Gender related findings on the total number of accidents are also not justified because the share of male and female car drivers and motorcycle drivers are unknown. So, the question is: are male car drivers involved in more door accidents because they are less cautious or because they drive more?

However, the greater odds of serious injury for male motorcycle drivers require attention.

Remark 4. ’Due to their slower response times, female motorcyclists are more often involved in door crash accidents, but because of their lower speed, female motorcyclists are less likely to suffer grave injuries.’

It is not clear whether n own finding or not. It sounds good but lacks justification.  ’female motorcyclists are more often involved’: please present the share of female motorcycle users. ’because of their lower speed’ please present data about travel speed of male and female motorcycle users.

Remark 5. Missing year in line 393 and 397. ‘Johnson et al.’

Remark 6. ’As for the type of motorcycles, in the long-term period, heavy motorcycles accounted for 74.1% of the door crashes as compared with only 35.3% of the light motorcycles.’

Since the share of motorcycle types is unknown, these findings may mislead the reader and cause a false perception that heavy motorcycle users are less cautious. However, if the authors present information on the share of motorcycle types, these findings are also valuable.

Remark 7. The analysis lacks data about speed and use of safety equipment, such as helmet and vests. How does it influence the findings?

Remark 8. It may be surprising that night-time is less dangerous than daytime. It may be good to compare this finding with similar papers. E.g., https://doi.org/10.3390/su13126945

Remark 9. The conclusion may focus on the own contribution. Therefore, the first two sentences of the conclusion may be deleted.

Remark 10. The ready-to-get-out indicator sounds good. However, this solution is derived from the difference between left and right side. How can the other findings generate specific safety measures? E.g., the difference between seasons.

Remark 11. ‘To the best of our knowledge, there has so far been no research or studies on motor-53 cycle-car door crashes in the world.’

The authors may provide a more comprehensive literature review because:

https://doi.org/10.1371/journal.pone.0208016

Reviewer 2 Report

The paper is interesting. I have only two major comments:

- The authors often refer to the previous similar analyses and contrast the findings. However, I am not sure if they took into account all the potential differences. For example, were the previous studies also based on court verdicts? (Probably not, as indicated on lines 395-397.) Obviously this may be one of the reasons of different findings.

- I would not use separate Section 5, but would merge it with Conclusions.

Additional minor comments:

- Some default text was kept on lines 149-156.

- In the list of references, journal name is wrongly labelled as "Accident; analysis and prevention" several times. The semicolon should be omitted.

Author Response

Dear Reviewer,     

I appreciate you giving your precious time to review the paper and for providing valuable comments. It was your valuable and insightful comments that led to possible improvements in the current version. The author has carefully considered the comments and tried my best to address every one of them. I hope the manuscript after careful revisions meet your high standards. The author welcomes further constructive comments if any.     

Below I provide the point-by-point responses. All modifications in the manuscript have been marked up using the “Track Changes” function.

Sincerely,

The author.

Point 1: The authors often refer to the previous similar analyses and contrast the findings. However, I am not sure if they took into account all the potential differences. For example, were the previous studies also based on court verdicts? (Probably not, as indicated on lines 395-397.) Obviously this may be one of the reasons of different findings.

Response 1: Thank you for your kind comments. I agree with your opinion. In the related literature reviews I found that there were differences between the two documents. The findings in this study can explain the differences, thus they were mentioned and explained. If you feel that it’s not appropriate here, I have deleted the two paragraphs as follows:

This result also explains the differences in gender in Pei’s [6] and Johnson et al.’s [7] studies. As Pei's 2011 study used the British Stats19 national road accident database, and the data extracted was from door crash accidents, women were more likely to be involved in door crash accidents. Police reports of injury crashes and hospital emergency department data were the primary sources of information for Johnson et al. (2013), and therefore men were more likely to be registered as getting injured during door crashes.

This also explains the differences regarding gender in the two British and Australian papers on bicycle-car door crash accidents.

Point 2: I would not use separate Section 5, but would merge it with Conclusions.

Response 2: Thank you for your kind comments. I have merged Section 5 and Conclusions.

Point 3: Some default text was kept on lines 149-156.

Response 3: Thank you for your kind comments. The instructions for authors between line 149 and 156 have been removed.

Point 4: In the list of references, journal name is wrongly labelled as "Accident; analysis and prevention" several times. The semicolon should be omitted.

Response 4: Thank you for your kind comments. The semicolon has been removed.

Reviewer 3 Report

Authors analyzes the risk factors of car door crashes. In my opinion, quite general information is provided in Introduction, there is a lack of specific research results, that have scientific value.

The author summarized and analyzed the collected information about such accidents. But this is more statistical information that is quite similar to other types of accidents. Analysis of results and discussion are more descriptive. The proposed solutions and recommendations are generally known and do not require further investigation.

The author should highlight the scientific novelty and originality of the research presented in this article. What is the purpose of this research?

The data used in the study are quite old. Are they really in line with the current situation?

In my opinion, Figures 1 and 4 are not typical for a scientific publication.

Author Response

Dear Reviewer,     

I appreciate you giving your precious time to review the paper and for providing valuable comments. It was your valuable and insightful comments that led to possible improvements in the current version. The author has carefully considered the comments and tried my best to address every one of them. I hope the manuscript after careful revisions meet your high standards. The author welcomes further constructive comments if any.     

Below I provide the point-by-point responses. All modifications in the manuscript have been marked up using the “Track Changes” function.

Sincerely,

The author.

Point 1: Authors analyzes the risk factors of car door crashes. In my opinion, quite general information is provided in Introduction, there is a lack of specific research results, that have scientific value. The author summarized and analyzed the collected information about such accidents. But this is more statistical information that is quite similar to other types of accidents. Analysis of results and discussion are more descriptive. The proposed solutions and recommendations are generally known and do not require further investigation.

Response 1: Thank you for your kind comments. Perhaps it is the different traffic conditions in the area the author investigated. When he was young, the author suffered a door crash accident. Fortunately, the speed was slow and there were no other vehicles behind, and he only suffered surface injuries. In 2010, when the author was already lecturing in university, there was a female student who was involved in the same type of accident on her way to the campus. She was not so lucky since after she was hit, she suffered from intracranial bleeding. Luckily an operation saved her life but afterwards, her intelligence and physical control were affected. Thus, the author began research in this area, and the risk factors for door crash accidents were obtained through investigations as a reference for accident prevention.     

Point 2: The author should highlight the scientific novelty and originality of the research presented in this article. What is the purpose of this research?

Response 2: Thank you for your kind comments. This research proposes recommendations for accident prevention through risk management using education, law enforcement and engineering, including the concept of the ready-to-get-out indicator using human factor engineering design. This light would alert motorcyclists to the cars parked alongside the road that there are drivers or passengers who may be about to get out of the car. They can then avoid the car or slow down to pass. The main purpose is to garner the attention of the government and automobile manufacturers through the public publication of this research and to attach importance to door crash accidents. I hope that you can also support the public publication of this research.

Point 3: The data used in the study are quite old. Are they really in line with the current situation?

Response 3: Thank you for your kind comments. Please forgive me, because the website from which I gathered the data was revised in 2017 due to data privacy issues, the public content of the data was adjusted. Hence, I was not able to retrieve the latest figures. However, I believe that this does not have a great impact upon the statistical results.

Point 4: In my opinion, Figures 1 and 4 are not typical for a scientific publication.

Response 4: Thank you for your kind comments. Figures 1 and 4 will be removed from the text, and the figure numbers will be revised.

Round 2

Reviewer 2 Report

Thank you for the revision. I can see that the review comments were addressed and the paper quality has improved. I recommend the paper to be accepted for publication in the journal.

Author Response

Thank you for your kind comment.

Reviewer 3 Report

Thanks to the author for corrections and comments. But I think that the amendments to the article do not significantly change the content of the article. The author's responses to the comments are more general, based on personal experience, but I wanted to draw the author's attention to the scientific novelty and originality of the article, which in my opinion is not sufficient. The author should also evaluate more recent data of such accidents.

Round 3

Reviewer 3 Report

Thanks to the author for making substantial corrections to the article and updating the data. I think the scientific novelty and originality of the article could be highlighted more.

Author Response

Thank you for your kind comment.